# Private Learning Implies Online Learning: An Efficient Reduction

**Alon Gonen**
University of California San Diego
algonen@cs.ucsd.edu

**Elad Hazan**
Princeton University and Google AI Princeton
ehazan@princeton.edu

**Shay Moran**
Google AI Princeton
shaymoran1@gmail.com

## Abstract

We study the relationship between the notions of differentially private learning and online learning in games. Several recent works have shown that differentially private learning implies online learning, but an open problem of Neel, Roth, and Wu [27] asks whether this implication is *efficient*. Specifically, does an efficient differentially private learner imply an efficient online learner?

In this paper we resolve this open question in the context of pure differential privacy. We derive an efficient black-box reduction from differentially private learning to online learning from expert advice.

## 1  Introduction

*Differential Private Learning* and *Online Learning* are two well-studied areas in machine learning. While at a first glance these two subjects may seem disparate, recent works gathered a growing amount of evidence which suggests otherwise. For example, *Adaptive Data Analysis* [15, 14, 24, 19, 3] shares strong similarities with adversarial frameworks studied in online learning, and on the other hand exploits ideas and tools from differential privacy. A more formal relation between private and online learning is manifested by the following fact:

*Every privately learnable class is online learnable*.

This implication and variants of it were derived by several recent works [20, 9, 1] (see the related work section for more details). One caveat of the latter results is that they are non-constructive: they show that every privately learnable class has a finite *Littlestone dimension*. Then, since the Littlestone dimension is known to capture online learnability [26, 5], it follows that privately learnable classes are indeed online learnable. Consequently, the implied online learner is not necessarily *efficient*, even if the assumed private learner is. Thus, the following question emerges:

Does efficient differentially private learning imply efficient online learning?

This question was explicitly raised by Neel, Roth and Wu [27]. In this work we resolve this question affirmatively under the assumption that the given private learner satisfies *Pure Differential Privacy* (the case of *Approximate Differential Privacy* remains open: see Section 4 for a short discussion). We give an efficient black-box reduction which transforms an efficient pure private learner to an efficient online learner. Our reduction exploits a characterization of private learning due to [4], together with tools from online boosting [6], and a lemma which converts oblivious online learning to adaptive online learning. The latter lemma is novel and may be of independent interest.

## 1.1 Main result

**Theorem 1.** *Let $\mathcal{A}$ be a differentially private learning algorithm for an hypothesis class $\mathcal{H}$ in the realizable setting. Denote its sample complexity by $m(\cdot, \cdot)$ and denote by $m_0 := m(1/4, 1/2)$. Then, Algorithm 3 is an efficient online learner for $\mathcal{H}$ in the realizable setting which attains an expected regret of at most $O(\sqrt{\ln(T)})$.*

The (standard) notation used in the theorem statement is detailed in Section 2.

**Agnostic versus Realizable.** It is natural to ask whether Theorem 1 can be generalized to the agnostic setting, namely, whether Algorithm 3 can be extended to an (efficient) online learner which achieves a sublinear regret against arbitrary adversaries. It turns out, that the answer is no, at least if one is willing to assume certain customary complexity theoretical assumptions and consider a non-uniform[1] model of computation. Specifically, consider the class of all halfspaces over the domain $\{0,1\}^n \subseteq \mathbb{R}^n$ whose margin is at least $\text{poly}(n)$. This class satisfies: (i) it is efficiently learnable by a pure differentially private algorithm [7, 18, 28]. (ii) Conditioned on certain average case hardness assumptions, there is no efficient online learner[2] for this class which achieves sublinear regret against arbitrary adversaries [11]. We note that this argument only invalidates the possibility of reducing agnostic online learning to realizable private learning. The question of whether there exists an efficient reduction from agnostic online learning to *agnostic* private learning remains open.

**Proof overview.** Here is a short outline of the proof. A characterization of differentially private learning due to [4] implies that if $\mathcal{H}$ is privately learnable in the pure setting, then the *representation dimension* of $\mathcal{H}$ is finite. Roughly, this means that for any fixed distribution $\mathcal{D}$ over labeled examples, by repeatedly sampling the (random) outputs of the algorithm $\mathcal{A}$ on a "dummy" input sample, we eventually get an hypothesis that performs well with respect to $\mathcal{D}$. In more detail, if one samples (roughly) $\exp(1/\alpha)$ random hypotheses, then with high probability one of them will have excess population loss $\leq \alpha$ with respect to $\mathcal{D}$. This suggests the following approach: sample $\exp(1/\alpha)$ random hypotheses ($\alpha$ will be specified later) and treat them an a class of experts, denoted by $\mathcal{H}_\alpha$; then, use *Multiplicative Weights* to online learn $\mathcal{H}_\alpha$ with regret (roughly) $\sqrt{T \log|H_\alpha|} \approx \sqrt{T/\alpha}$, and thus the total regret will be

$$\alpha \cdot T + \sqrt{T/\alpha},$$

which is at most $T^{2/3}$ if we set $\alpha = T^{-1/3}$.

There are two caveats with this approach: i) the number of experts in $H_\alpha$ is $\exp(T^{1/3})$, which is too large for applying Multiplicative Weights efficiently. ii) A more subtle issue is that the above regret analysis only applies in the *oblivious* setting: an adaptive adversary may "learn" the random class $\mathcal{H}_\alpha$ from the responses of our online learner, and eventually produce a (non-typical) sequence of examples for which it is no longer the case that the best expert in $\mathcal{H}_\alpha$ has loss $\leq \alpha$. To handle the first obstacle we only require a constant accuracy of $\alpha = 1/4$, which we later reduce using online boosting from [6]. As for the second obstacle, to cope with adaptive adversaries we propose a general reduction from the adaptive setting which might be of independent interest.

## 1.2 Related work

**Online and private learning** Feldman and Xiao [20] exploited techniques from communication complexity to show that every pure differentially private (DP) learnable class has a finite Littlestone dimension (and hence is online learnable). Their work actually proved that *pure* private learning is strictly more difficult than online learning. That is, there exist classes with a finite Littlestone dimension which are not pure-DP learnable. More recently, Alon et al. [9, 1] extended the former result to approximate differential privacy, showing that every approximate-DP learnable class has a finite Littlestone dimension. It remains open whether the converse holds.

Another line of work by [27, 8] exploit online learning techniques to derive results in differential privacy related to *sanitization* and *uniform convergence*.

**Adaptive data analysis.** A growing area which intersects both fields of online learning and private learning is *adaptive data analysis* ([15], [14],[24] [19],[3]). This framework studies scenarios in which a data analyst wishes to test multiple hypotheses on a finite sample in an adaptive manner. The adaptive nature of this setting resembles scenarios that are traditionally studied in online learning, and the connection with differential privacy is manifested in the technical tools used to study adaptive data analysis, many of which were developed in differential privacy (e.g. composition theorems).

**Oracle complexity of online learning.** One feature of our algorithm is that it uses an oracle access to a private learner. Several works studied online learning in oracle model ([23, 25, 13]). This framework is natural in scenarios in which it is computationally hard to achieve sublinear regret in the worst case, but the online learner has access to an offline optimization and/or learning oracle. Our results fall into the same paradigm, where the oracle is a differentially private learner.

## 2 Definitions and Preliminaries

### 2.1 PAC learning

Let $\mathcal{X}$ be an *instance space*, $\mathcal{Y} = \{-1, 1\}$ be a *label set*, and let $\mathcal{D}$ be an (unknown) distribution over $\mathcal{X} \times \mathcal{Y}$. An "$\mathcal{X} \to \mathcal{Y}$" function is called a concept/hypothesis. The goal here is to design a learning algorithm, which given a large enough input sample $S = ((x_1, y_1)), \ldots, (x_m, y_m))$ drawn i.i.d. from $\mathcal{D}$, outputs an hypothesis $h : \mathcal{X} \to \mathcal{Y}$ whose *expected risk*

$$L_{\mathcal{D}}(h) := \mathbb{E}_{(x,y) \sim \mathcal{D}}[\ell(h(x), y)] \qquad \ell(a, b) = \mathbf{1}_{a \neq b}$$

is small compared to the best hypothesis in a *hypothesis class* $\mathcal{H}$, which is a fixed and known to the algorithm.

The distribution $\mathcal{D}$ is said to be realizable with respect to $\mathcal{H}$ if there exists $h^{\star} \in \mathcal{H}$ such that $L_{\mathcal{D}}(h) = 0$. We also define the empirical risk of an hypothesis $h$ with respect to a sample $S = ((x_1, y_1), \ldots, (x_m, y_m))$ as $L_S(h) = \frac{1}{m} \sum_{i=1}^{m} \ell(h(x_i), y_i)$.

**Definition 2. (PAC learning)** An hypothesis class $\mathcal{H}$ is PAC learnable with sample complexity $m(\alpha, \beta)$ if there exists an algorithm $\mathcal{A}$ such that for any distribution $\mathcal{D}$ over $\mathcal{X}$, an accuracy and confidence parameters $\alpha, \beta \in (0, 1)$, if $\mathcal{A}$ is given an input sample $S = ((x_1, y_m), \ldots, (x_m, y_m)) \sim \mathcal{D}^m$ such that $m \geq m(\alpha, \beta)$, then it outputs an hypothesis $h : \mathcal{X} \to \mathcal{Y}$ satisfying $L_{\mathcal{D}}(h) \leq \alpha$ with probability at least $1 - \beta$. The class $\mathcal{H}$ is efficiently PAC learnable if the runtime of $\mathcal{A}$ (and thus its sample complexity) are polynomial in $1/\alpha$ and $1/\beta$. If the above holds only for realizable distributions then we say that $\mathcal{H}$ is PAC learnable in the realizable setting.

### 2.2 Differentially private PAC learning

In some important learning tasks (e.g. medical analysis, social networks, financial records, etc.) the input sample consists of sensitive data that should be kept private. Differential privacy ([12, 16]) is a by-now standard formalism that captures such requirements.

The definition of differentially private algorithms is as follows. Two samples $S', S'' \in (\mathcal{X} \times \mathcal{Y})^m$ are called *neighbors* if there exists at most one $i \in [m]$ such that the $i$'th example in $S'$ differs from the $i$'th example in $S''$.

**Definition 3. (Differentially private learning)** A learning algorithm $\mathcal{A}$ is said to be $\epsilon$-differentially private[3] (DP) if for any two neighboring samples and for any measurable subset $\mathcal{F} \in \mathcal{Y}^{\mathcal{X}}$,

$$Pr[\mathcal{A}(S) \in \mathcal{F}] \leq \exp(\epsilon) Pr[\mathcal{A}(S') \in \mathcal{F}] \text{ and}$$
$$Pr[\mathcal{A}(S') \in \mathcal{F}] \leq \exp(\epsilon) Pr[\mathcal{A}(S) \in \mathcal{F}]$$

*Group privacy* is a simple extension of the above definition [17]: Two samples $S, S'$ are $q$-neighbors if they differ in at most $q$ of their pairs.

**Lemma 4.** *Let $\mathcal{A}$ be a DP learner. Then for any $q \in \mathbb{N}$ and any two $q$-neighboring samples $S, S'$ and any subset $\mathcal{F} \in \mathcal{Y}^{\mathcal{X}} \cap \mathrm{range}(\mathcal{A})$, $Pr[\mathcal{A}(S) \in \mathcal{F}] \leq \exp(\epsilon q) Pr[\mathcal{A}(S') \in \mathcal{F}]$*

Combining the requirements of PAC and DP learnability yields the definition of private PAC (PPAC) learner.

**Definition 5. (PPAC Learning)** A concept class $\mathcal{H}$ is differentially private PAC learnable with sample complexity $m(\alpha, \beta)$ if it is PAC learnable with sample complexity $m(\alpha, \beta)$ by an algorithm $\mathcal{A}$ which is an $\epsilon = 0.1$-differentially private.

**Remark.** Setting $\epsilon = 0.1$ is without loss of generality; the reason is that there are efficient methods to boost the value of $\epsilon$ to arbitrarily small constants, see [30] and references within.

## 2.3 Online Learning

The online model can be seen as a repeated game between a learner $\mathcal{A}$ and an environment (a.k.a. adversary) $\mathcal{E}$. Let $T$ be a (known[4]) horizon parameter. On each round $t \in [T]$ the adversary decides on a pair $(x_t, y_t) \in \mathcal{X} \times \mathcal{Y}$, and the learner decides on a prediction rule $h_t : \mathcal{X} \to \{0, 1\}$. Then, the learner suffers the loss $|y_t - \hat{y}_t|$, where $\hat{y}_t = h(x_t)$. Both players may base their decisions on the entire history and may use randomness. Unlike in the statistical setting, the adversary $\mathcal{E}$ can generate the examples in an adaptive manner. In this work we focus on the *realizable* setting where it is assumed that the labels are realized by some target concept $c \in \mathcal{H}$, i.e., for all $t \in [T]$, $y_t = c(x_t)$.[5] The measure of success is the expected number of mistakes done by the learner:

$$\mathbb{E}[M_{\mathcal{A}}] = \mathbb{E}\Big[\sum_{t=1}^{T} \ell(\hat{y}_t, y_t)\Big],$$

where the expectation is taken over the randomness of the learner and the adversary. An algorithm $\mathcal{A}$ is a (strong) online learner if for any horizon parameter $T$ and any realizable sequence $((x_1, y_1), \ldots, (x_T, y_T))$, the expected number of mistakes made by $\mathcal{A}$ is sublinear in $T$.

### 2.3.1 Weak Online Learning

We describe an extension due to [6] of the boosting framework ([29]) (from the statistical setting) to the online.

**Definition 6. (Weak online learning)** An online learner $\mathcal{A}$ is called a weak online learner for a class $\mathcal{H}$ with an *edge* parameter $\gamma \in (0, 1/2)$ and *excess loss* parameter $T_0 > 0$ if for any horizon parameter $T$ and every sequence $((x_1, y_1), \ldots, (x_T, y_T))$ realized by some target concept $c \in \mathcal{H}$, the expected number of mistakes done by $\mathcal{A}$ satisfies

$$\mathbb{E}[M_{\mathcal{A}}] \leq \left(\frac{1}{2} - \gamma\right) T + T_0 \, .$$

### 2.3.2 Oblivious vs. Non-oblivious Adversaries

The general adversary considered in this paper is adaptive in the sense that it can choose the pair $(x_t, y_t)$ based on the actual predictions $\hat{y}_1, \ldots, \hat{y}_{t-1}$ made by the learner on rounds $1, \ldots, t-1$. An adversary is called *oblivious* if it chooses the entire sequence $((x_1, y_1), \ldots, (x_T, y_T))$ in advance. We will first develop an online weak online learner for the oblivious setting and then extend it to the adaptive setting.

### 2.3.3 Regret bounds using Multiplicative Weights

Although we focus our attention on the realizable setting, our development also requires working in the so-called agnostic setting, where the sequence $((x_1, y_1), \ldots, (x_T, y_T))$ is not assumed to be

realized by some $c \in \mathcal{H}$. The standard measure of success in this setting is the expected *regret* defined as

$$\mathbb{E}[\mathrm{Regret}_T] = \mathbb{E} \sum_{t=1}^{T} \ell(\hat{y}_t, y_t) - \inf_{h \in \mathcal{H}} \sum_{t=1}^{T} \ell(h(x_t), y_t).$$

Accordingly, an online learner in this context needs to achieve a sublinear regret in terms of the horizon parameter $T$.

When the class $\mathcal{H}$ is finite, there is a well-known algorithm named *Multiplicative Weights* (MW) which maintains a weight $w_{t,j}$ for each hypothesis (a.k.a. *expert* in the online model) $h_j$ according to

$$w_{1,j} = 1 , \quad w_{t+1,j} = w_{t,j} \exp(-\eta \ell(h_j(x_t), y_t)))$$

where $\eta > 0$ is a step-size parameter. At each time $t$, MW predicts with $\hat{y}_t = h_j(x_t)$ with probability proportional to $w_{t,j}$. We refer to [2] for an extensive survey on Multiplicative Weights and its many applications. The following theorem establishes an upper bound on the regret of MW.

**Theorem 7.** *(Regret of MW) If the class $\mathcal{H}$ is finite then the expected regret of MW with step size parameter $\eta = \sqrt{\log(|H|)/T}$ is at most $\sqrt{2T \log |H|}$.*

## 3 The Reduction and its Analysis

In this section we formally present our efficient reduction from online learning to private PAC learning. Our reduction only requires a black-box oracle access to the the private learner. The reduction can be roughly partitioned into 3 parts: (i) We first use this oracle to construct an efficient weak online learner against oblivious adversaries. (ii) Then, we transform this learner so it also handles adaptive adversaries. This step is based on a general reduction which may be of independent interest. (iii) Finally, we boost the weak online learner to a strong one using online boosting.

### 3.1 A Weak Online Learner in the Oblivious Setting

Let $\mathcal{A}_p$ be a PPAC algorithm with sample complexity $m(\alpha, \beta)$ for $\mathcal{H}$ and denote by $m_0 := m(1/4, 1/2) = \Theta(1)$. We only assume an oracle access to $\mathcal{A}_p$, and in the first part we use it to construct a distribution over hypotheses/experts. Specifically, let $S_0$ be a dummy sample consisting of $m$ occurrences of the pair $(\bar{x}, 0)$ where $\bar{x}$ is an arbitrary instance from $\mathcal{X}$. Note that the hypothesis/expert $\mathcal{A}_p(S_0)$ is random.[6]

**Definition 8.** Let $P_0$ be the distribution over hypotheses/experts induced by applying $\mathcal{A}_p$ on the input sample $S_0$.

**Lemma 9.** *For any realizable distribution $\mathcal{D}$ over $\mathcal{X} \times \mathcal{Y}$, with probability at least $15/16$ over the draw of $N = \Theta(\exp(m_0)) = \Theta(1)$ i.i.d. hypothesis $h_1, \dots, h_N \sim P_0$ , there exists $i \in [N]$ such that $L_{\mathcal{D}}(h_i) \leq 1/4$.*

*Proof.* Let $c \in \mathcal{H}$ be such that $L_{\mathcal{D}}(c) = 0$, and denote by

$$\mathcal{H}(\mathcal{D}) = \{h \in \mathrm{range}(\mathcal{A}) : \ L_{\mathcal{D}}(h) \leq 1/4\} .$$

By assumption, if we feed the PPAC algorithm $\mathcal{A}$ with a sample $S$ drawn according to $\mathcal{D}^m$ and labeled by $c$, then with probability at least $1/4$ over both the internal randomness of $\mathcal{A}$ and the draw of $S$, the output of $\mathcal{A}$ belongs to $\mathcal{H}(\mathcal{D})$. It follows that there exists at least one sample, which we denote by $\bar{S}$, such that with probability at least $1/2$ over the randomness of $\mathcal{A}$, the output $h = \mathcal{A}(\bar{S})$ belongs to $\mathcal{H}(\mathcal{D})$. Since $\mathcal{A}$ is differentially private and $(\bar{S}, S_0)$ are $m$-neighbors, we obtain that

$$Pr[\mathcal{A}(S_0) \in \mathcal{H}(\mathcal{D})] \geq \frac{1}{2} \exp(-0.1m_0) .$$

Consequently if we draw $N = \Theta(\exp(m_0))$ hypotheses $h_j \sim P_0$ then with probability at least $15/16$, at least one of the $h_j$'s belongs to $\mathcal{H}(\mathcal{D})$. This completes the proof. $\square$

Armed with this lemma, we proceed by applying the Multiplicative Weights method to the random class $H$ produced by the PPAC learner $\mathcal{A}_p$. The algorithm is detailed as Algorithm 1. The next lemma establishes its weak learnability in the oblivious setting.

**Algorithm 1** Weak online learner for oblivious adversaries

---

**Oracle access:** Let $P_0$ denote the distribution from Definition 8, and let $m_0 = m(1/4, 1/2)$, where $m(\alpha, \beta)$ is the sample complexity of the private learner $\mathcal{A}_p$.

**Set**: $N = \Theta(\exp(m_0))$, $\eta = \sqrt{\frac{\log N}{T}}$.

**for** $j = 1$ to $N$ **do**
    $h_j \sim P_0$
    $w_{1,j} = 1$     $\forall j \in [N]$                         ▷ Initializing MW w.r.t. $h_1, \ldots, h_N$
**end for**
**for** $t = 1$ to $T$ **do**
    Receive an instance $x_t$
    Predict $\hat{y}_t = h_j(x_t)$ with probability $w_{t,j} / \sum_{k=1}^{N} w_{t,k}$
    Receive the true label $y_t$
    $w_{t+1,j} = w_{t,j} \exp(-\eta |y_t - h_j(x_t)|)$
**end for**

---

**Lemma 10.** *For any oblivious adversary and horizon parameter $T$, the expected number of mistakes made by Algorithm 1 is at most $O\left(\sqrt{T m_0} + \frac{T}{4}\right)$. In particular, the algorithm is a weak online learner with an edge parameter $1/8$ and excess loss $T_0 = O(1)$.*

*Proof.* Since the adversary is oblivious, it chooses the (realizable) sequence $(x_1, y_1) \ldots, (x_T, y_T)$ in advance. In particular, these choices do not depend on the hypotheses $h_1, \ldots, h_N$ drawn from $P_0$. Define a distribution $\mathcal{D}$ over $\mathcal{X} \times \mathcal{Y}$ by

$$\mathcal{D}[\{(x, y)\}] = \frac{|\{t \in [T] : (x_t, y_t) = (x, y)\}|}{T}.$$

By the previous lemma we have that with probability at least $15/16$, there exists $j \in [N]$ such that

$$\frac{1}{T} \sum_{t=1}^{T} \ell(h_j(x_t), y_t) = L_{\mathcal{D}}(h_i) \leq 1/4.$$

Using the standard regret bound of Multiplicative Weights (Lemma 7), we obtain that the expected number of mistakes done by our algorithm is at most

$$2\sqrt{T \log N} + \frac{T}{4} + T/16.$$

(The $T/16$ factor is because the success probability of $\mathcal{A}_p$ is $15/16$, see Lemma 9). In particular, set $T_0 = C \cdot \log N = O(m_0)$ for a sufficiently large constant $C$ such that,

$$2\sqrt{T \log N} + \frac{T}{4} + \frac{T}{16} \leq \left(\frac{1}{2} - \frac{1}{8}\right) T + T_0.$$

This concludes the proof.          □

### 3.2 General reduction from adaptive to oblivious environments

In this part we describe a simple general-purpose extension from the oblivious setting to the adaptive setting. Let $\mathcal{A}_o$ be an online learner for $\mathcal{H}$ that handles oblivious adversaries. We may assume that $\mathcal{A}_o$ is random since otherwise any guarantee with respect to oblivious adversary holds also with respect to adaptive adversary. Given an horizon parameter $T$, we initialize $T$ instances of this algorithm (each of with an independent random seed of its own). Finally, on round $t$ we follow the prediction of the $t$-th instance, $\mathcal{A}_o^{(t)}$.

**Lemma 11.** *Suppose that $\mathcal{A}_o$ is an online learner for a class $\mathcal{H}$ in the oblivious setting whose expected regret is upper bounded by $R(T)$. Then, the expected regret of Algorithm 2 is also upper bounded by $R(T)$.*

*Proof.* The proof relies on a lemma by [10] which provides a reduction from the adaptive to the oblivious setting given a certain condition on the responses of the online learner. Since this lemma

---

**Algorithm 2** Reduction from Oblivious to Adaptive Setting

---

    **Oracle access:** Online algorithm $\mathcal{A}_o$ for the oblivious setting.

    **Initialize** $T$ independent instances of $A_o$, denoted $A_o^{(1)}, \ldots, A_o^{(T)}$.

    **for** $t = 1$ to $T$ **do**

        $\hat{y}_t^{(j)} :=$ prediction of $\mathcal{A}_o^{(j)}$, $j = 1, \ldots, T$.

        Predict $\hat{y}_t = \hat{y}_t^{(t)}$

    **end for**

---

is somewhat technical, we defer the proof of the stated bound to the appendix (Section A), and prove here a slightly weaker bound, which is off by a factor of $\log T$. This weaker bound however follows from elementary arguments in a self contained manner.

Note that the algorithms $A^{(j)}$'s for $j = 1 \ldots T$ are i.i.d. (i.e. have independent internal randomness). Therefore, *the sequence of examples chosen by the adversary up to time $t$ is independent of the predictions of $A_o^{(j)}$ whenever $j \geq t$*, and thus we can use the assumed guarantee for $A_o^{(j)}$ in the oblivious setting:

$$(\forall j \geq t) : \mathbb{E}\Big[\sum_{i=1}^{t} \hat{\ell}_i^{(j)}\Big] \leq R(T), \tag{1}$$

where $\hat{\ell}_i^{(j)} = \ell(y_i, \hat{y}_i^{(j)})$. Similarly, it follows that

$$\mathbb{E}[\hat{\ell}_t] = \mathbb{E}[\hat{\ell}_t^{(t)}] = \mathbb{E}[\hat{\ell}_t^{(t+1)}] = \ldots = \mathbb{E}[\hat{\ell}_t^{(T)}] = \mathbb{E}\left[\frac{1}{T-t+1}\sum_{j=t}^{T}\hat{\ell}_t^{(j)}\right]. \tag{2}$$

Therefore,

$$\mathbb{E}\Big[\sum_{t=1}^{T}\hat{\ell}_t\Big] = \mathbb{E}\Big[\sum_{t=1}^{T}\frac{1}{T-t+1}\sum_{j=t}^{T}\hat{\ell}_t^{(j)}\Big] \qquad \text{(by Equation 2)}$$

$$= \mathbb{E}\Big[\sum_{j=1}^{T}\sum_{t=1}^{j}\frac{\hat{\ell}_t^{(j)}}{T-t+1}\Big]$$

$$\leq \mathbb{E}\Big[\sum_{j=1}^{T}\sum_{t=1}^{j}\frac{\hat{\ell}_t^{(j)}}{T-j+1}\Big]$$

$$= \sum_{j=1}^{T}\frac{\mathbb{E}[\sum_{t=1}^{j}\hat{\ell}_t^{(j)}]}{T-j+1}$$

$$\leq \sum_{j=1}^{T}\frac{R(T)}{T-j+1} \qquad \text{(by Equation 1)}$$

$$\leq R(T)\log T.$$

$\square$

### 3.3   Applying Online Boosting

In this part we apply an online boosting algorithm due to [6] to improve the accuracy of our weak learner. The algorithm is named Online Boosting-by-Majority (online BBM). We start by briefly describing online BBM and stating an upper bound on its expected regret.

The Online BBM can be seen as an extension of Boosting-by-Majority algorithm due to [21]. Let WL be a weak learner with an edge parameter $\gamma \in (0, 1/2)$ and excessive loss $T_0$. The online BBM algorithm maintains $N$ copies WL, denoted by $\text{WL}^{(1)}, \ldots, \text{WL}^{(N)}$. On each round $t$ it uses a simple

(unweighted) majority vote over $\texttt{WL}^{(1)}, \ldots, \texttt{WL}^{(N)}$ to perform a prediction $\hat{y}_t$. The pair $(x_t, y_t)$ is passed to the weak learner $\texttt{WL}^{(j)}$ with probability that depends on the accuracy of the majority vote based on the weak learners $\texttt{WL}^{(1)}, \ldots, \texttt{WL}^{(j-1)}$ with respect to $(x_t, y_t)$. Similarly to the well-known AdaBoost algorithm by [22], the worse is the accuracy of the previous weak learners, the larger is the probability that $(x_t, y_t)$ is passed to $\texttt{WL}^{(j)}$ (see Algorithm 1 in [6]).

**Theorem 12.** *([6]) For any $T$ and any $N$, the expected number of mistakes made by the Online Boosting-by-Majority Algorithm is bounded by[7]*

$$\exp\left(-\frac{1}{2}N\gamma^2\right)T + \tilde{O}\left(\sqrt{N}(T_0 + \frac{1}{\gamma})\right)$$

*In particular, if $\gamma$ and $T_0$ are constants then for any $\epsilon > 0$, it suffices to pick $N = \Theta(\ln(1/\epsilon))$ weak learners to obtain an upper bound of*

$$O(T\epsilon + \sqrt{\ln(1/\epsilon)}) \tag{3}$$

*on the expected number of mistakes.*

We have collected all the pieces of our algorithm.

---

**Algorithm 3** Online Learning using a Private Oracle
___

  **Horizon parameter:** $T$
  $\epsilon := 1/T$
  Weak learner $\texttt{WL}$: Algorithm 2 applied to Algorithm 1
  Apply online BBM using $N = \Theta(\ln(1/\epsilon)) = \Theta(\ln T)$ instances of $\texttt{WL}$

---

*Proof.* **(of Theorem 1)** Combining Lemma 10 and Lemma 11, we obtain that $\texttt{WL}$ is a weak online learner with an edge parameter $\gamma = 1/8$ and constant excessive loss. Plugging $\epsilon = 1/T$ in the accuracy parameter in Theorem 12 (Equation 3) yields the stated bound. □

## 4  Discussion

We have considered online learning in the presence of a private learning oracle, and gave an efficient reduction from online learning to private learning.

We conclude with two questions for future research.

- Can our result can be extended to the approximate case? That is, does an efficient approximately differentially private learner for a class $\mathcal{H}$ imply an efficient online algorithm with sublinear regret? Can the online learner be derived using only an oracle access to the private learner?

- Can our result be extended to the agnostic setting? That is, does an efficient agnostic private learner for a class $\mathcal{H}$ implies an efficient agnostic online learner for it?

As for the first question, one difference with pure privacy is that Lemma 9 ceases to hold. Recall that this lemma guarantees that applying a pure private learner on a dummy sample yields an output hypothesis, which is correlated with any realizable target concept, with a non-negligible chance. This lemma manifests a certain obliviousness of pure private learners which is crucial in our transformation from the statistical i.i.d setting to the adversarial setting. Approximately private learners do not share similar obliviousness: in particular, to obtain an output hypothesis which is non-trivially correlated with the realizable target concept, one must apply it on an i.i.d sample consistent with the concept (rather than an arbitrary dummy sample).

As for the second question, it is natural to try and implement the approach used in this paper, but there are several major missing components. Most notably, we would need a boosting algorithm for agnostic online learning, which is not known to exist. We consider this as a direction for future research, which is interesting in its own right.

## Footnotes

[1]Complexity theory distinguishes between uniform and non-uniform models, such as Turing machines vs. arithmetic circuits. In this paper we consider the uniform model. However, the lower bound we sketch applies to non-uniform computation.

[2]The result in [11] is in fact stronger: it shows that there exists no efficient agnostic PAC learner for this class (see Theorem 1.4 in it).

[3]The algorithm is said to be $(\epsilon, \delta)$-approximate differentially private if the above inequality holds up to an additive factor $\delta$. In this work we focus on the so-called pure case where $\delta = 0$.

[4]Standard doubling techniques allow the learner to cope with scenarios where $T$ is not known.

[5]However, the adversary does not need to decide on the identity of $c$ in advance.

[6]The definition of differential privacy implies that every private algorithm is randomized (ignoring trivialities).

[7]The bound in [6] is an high probability bound. It is easy to translate it to a bound in expectation.

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

## A  Proof of Lemma 11

The proof exploits Lemma 4.1 from [10] which we explain next. Let $\mathcal{A}$ be a (possibly randomized) online learner, and let $u_t$ denote the response of $\mathcal{A}$ in time $t \leq T$. Then, since $\mathcal{A}$ may be randomized, $u_t$ is drawn from a random variable $U_t$ that may depend on the entire history: namely, on both the responses of $\mathcal{A}$ as well as of the adversary up to time $t$. So

$$U_t = U_t(u_1 \ldots u_{t-1}, v_1 \ldots v_{t-1}),$$

where $u_i \sim U_i$ denotes the response of $\mathcal{A}$ and $v_i \sim V_i$ denotes the response of the (possibly randomized) adversary on round $i < t$ (in the classifications setting, $v_i$ is the labelled example $(x_i, y_i)$, and $u_i$ is the prediction rule $h_i : \mathcal{X} \to \{0,1\}$ used by $\mathcal{A}$). Lemma 4.1 in [10] asserts that if $U_t$ is only a function of the $v_i$'s, namely

$$U_t = U_t(v_1 \ldots v_{t-1}), \tag{4}$$

then the expected regret of $\mathcal{A}$ in the adaptive setting is the same like in the oblivious setting.

The proof now follows by noticing that Algorithm 2 satisfies Equation (4). To see this, note that at each round $t$, Algorithm 2 uses the response of algorithm $A_o^{(t)}$ which only *depends on the responses*

*of the adversary and $A_o^{(t)}$ up to time $t$.* In particular, it does not additionally depend the responses of Algorithm 2 at times up to $t$. Putting it differently, given the responses of the adversary $z_1 \ldots z_{t-1}$, one can produce the response of Algorithm 2 at time $t$ by simulating $A_o^{(t)}$ on this sequence.

Thus, we may assume that the adversary is oblivious, and therefore that the sequence of examples $(x_1, y_1) \ldots (x_t, y_t)$ is fixed in advance and independent from the algorithms $A_o^j$'s. Now, since $A_o^{(1)}, \ldots, A_o^{(T)}$ are i.i.d. (i.e. have independent internal randomness), the expected loss of Algorithm 2 at time $t$ satisfies

$$\mathbb{E}[\hat{\ell}_t] = \mathbb{E}[\hat{\ell}_t^{(t)}] = \mathbb{E}[\hat{\ell}_t^{(1)}] = \ldots = \mathbb{E}[\hat{\ell}_t^{(T)}] = \mathbb{E}\left[\frac{1}{T}\sum_{j=1}^{T}\hat{\ell}_t^{(j)}\right],$$

where $\hat{\ell}_i^{(j)} = \ell(y_i, \hat{y}_i^{(j)})$. Thus, its expected number of mistakes is at most

$$\mathbb{E}\left[\sum_{t=1}^{T}\hat{\ell}_t\right] = \mathbb{E}\left[\sum_{t=1}^{T}\frac{1}{T}\sum_{j=1}^{T}\hat{\ell}_t^{(j)}\right] = \frac{1}{T}\sum_{j=1}^{T}\mathbb{E}\left[\sum_{t=1}^{T}\hat{\ell}_t^{(j)}\right].$$

Therefore, the expected regret satisfies

$$\mathbb{E}[\text{Regret}_T] = \frac{1}{T}\sum_{j=1}^{T}\mathbb{E}[\text{Regret}_T^{(j)}] \leq R(T).$$

