[Reviews · NeurIPS 2019]

Reviewer 1



Summary: The goal of the paper is to show that, given a private PAC learner for a hypothesis class, one can do computationally efficient online no-regret learning with that class. The steps taken are: 1. Get a weak learner against oblivious adversaries as follows: sample the private PAC learner on a dummy dataset many times; use these hypotheses as experts with multiplicative weights. 2. Transform this to a weak learner against adaptive adversaries (cute, general reduction/observation) 3. Use online boosting to get a 'strong' learner from the weak one. Opinion: I think the paper's result is reasonably interesting and significant (though I am not an expert in this literature). The proofs are all clean and efficiently organized. On the other hand, the technical contribution is somewhat limited -- most of it is in step 1, the others are more like (nice!) observations. Originality: original results as far as I know (while studying a relatively standard setting and known open problem). Quality: Seems to be a high quality of writing and technical analysis. Clarity: High in my opinion. Significance: Medium in my opinion, perhaps not too high because of the relatively narrow focus and results. Comments: Line 2: suggest cutting "in games", doesn't seem accurate to the paper. Line 7-8: a bit confusing. I think you reduce the problems in the opposite direction stated. Or, you online learn by efficiently calling d.p. learning as a black box. Consider clarifying. Line 60: period should be a semicolon? Line 95-96: notation <= with squiggly should be defined, or use big-O How Def 6 interacts with the oblivious/non adversary definition is not clear. What is the expectation over in Def 6? The sequence must be fixed in advance, so presumably there is no "adversary" at all, but clarifying may be smart. Algorithm 1, first line: for some reason this references as "Def 3.1" but points to Def 8. Line 231: implies -> imply Please respond to any part of my review you think helpful. ---- Post-discussion: Thank you for the informative response. I agree that a bit of discussion on difficulty of extensions would be valuable.

Reviewer 2



This is a very succinct paper and I think that it serves as a nice theoretical contribution. One exciting feature is that the reduction builds on the online boosting framework of Beygelzimer, Kale, and Luo - to my knowledge this is the first theory result that uses their technique as a subroutine. The reduction from oblivious to adversarial adversaries seems new as well. One comment I have would be to spend a bit more time discussing *why* the techniques work. In particular, I found the result of Lemma 9 rather counterintuitive: It essentially says that if we run any pure DP pac learning algorithm a constant number of times on any arbitrary dataset, one of the outputs will have nontrivial classification accuracy on an unrelated target distribution with constant probability. If this hasn't been noted anywhere before it seems worth mentioning as a structural result. Some misc typos/comments: * Proof of lemma 9: $m$ and $m_{0}$ seem to be confused in various places/not defined. * 173: Shouldn't this be $m(1/4, 1/4)$ rather than $m(1/4, 1/2)$? * between line 181 and 182: Should this be "with probability at least 1-1/4" rather than "with probability at least 1/4"? * Where is $\mathcal{H}(\mathcal{D},c)$ defined?

Reviewer 3



Originality: This work resolves an open question. The approach is based on a characterization of private learning as well as recent progress in online learning and boosting. Quality: The technical claims in the paper are supported by valid proofs. While there might be slight inaccuracy/typo in the statement of lemmas and theorems (see Part 5 (1) and (2)), these issues can be easily fixed and do not affect the validity of results. Clarity: The paper is beautifully written and very easy to understand. In particular, Section 1.1 gives a clear overview of their proof and also suggests some technical issues that have to be addressed. The full proof is very structured and easy to follow. Significance: This work resolves an open question regarding the relation between online learning and differential privacy. While I am not an expert on these topics, this result seems solid and significant to me. *** added after author feedback *** I have read the author feedback and thank the authors for answering the questions.

[Author Response · NeurIPS 2019]

We thank the reviewers for their careful review and valuable suggestions. We will revise the paper accordingly. Below, we address some specific comments and questions.

# 1 Reviewer 1

Thank you for your helpful comments and suggestions, we will include them in the final version of the paper. We do have one main comment which is relevant to consider raising your score (point 1 below).

- "Significance: Medium in my opinion, perhaps not too high because of the relatively narrow focus and results...I don't think it could be improved much except by expanding the scope.":
  While this paper focuses on proving a single concrete statement, this statement bridges two well-researched areas within machine learning: *Online Learning* and *Private Learning*. Moreover, it is one of the only such papers which combines non-trivial techniques from both areas (e.g. the representation dimension and online boosting). As such, we believe it could be of interest to researchers from both areas and lead to more interaction between them. Consequently, the scope of this paper is actually wider than appears at first sight and spans two separate research communities.
- "Line 7-8: a bit confusing...": indeed the reduction is from online learning to private learning. We will fix it.
- "How Def 6 interacts with oblivious...": our proof needs weak online learners in both the oblivious and adaptive setting. We will clarify this in the definition.
- "...What is the expectation over in Def 6?...": the expectation in definition 6 is w.r.t. the randomness of both the learner and the adversary.

# 2 Reviewer 2

- "One comment I have would be...": We agree that Lemma 2 is counter-intuitive at a first sight. This is largely due to the fact that Pure Differential Privacy is a very strong requirement. More precisely, the notion of Differential Privacy requires the learner to be stable w.r.t. modification of one example in the sample. This enables replacing the entire sample and still getting a non-trivial correlation.
  This result is not new. As we mention in the proof overview (section 1.1), Lemma 9 is very close to a similar characterization shown in [4].
- "The most obvious improvement...to address either of the open problems...": see the last item in our reply to Reviewer 3.

# 3 Reviewer 3

- "Lemma 10: The big-O...": indeed, the $\log(T)$ term in lemma 10 is redundant.
- "Theorem 12: Could the $\ln(1/\epsilon)$ term in equation (1)...": indeed, the bound in Theorem 12 can be improved to $T\epsilon + \sqrt{\ln(1/\epsilon)}$. As a result, the dependence on $T$ in Theorem 1 can be improved to $\sqrt{\ln T}$. Thanks!
- Discussing possible directions for extending our results: an earlier draft of this paper had a longer and more detailed discussion of such possible directions and difficulties, but we decided to omit it as we felt that it was a bit too long and technical. As two reviewers expressed interest, we will integrate (at least some of) it back. Essentially, the main challenges are:
  - Extension to pure Differential Privacy: we don't know of any useful extension of the structural characterization of pure Differential Privacy (Lemma 9) to the approximate case.
  - We do not know of any (online) boosting algorithm for the agnostic setting whose guarantees are useful for our setting.

[Meta-Review · NeurIPS 2019]

A theory paper showing that a reduction from online learning to differentially private PAC learning, e.g., a differentially private PAC learner for concept class H can be used in a black-box fashion to obtain a low regret bound for online learning with 0/1 loss for concept class H. The key contribution here is that this is a computationally efficient reduction, which resolves an open problem of Neel/Roth/Wu and improves on the information theoretic result of Feldman/Xiao. In addition to the strong result, the reviewers point out that the paper contains many interesting observations along the way (e.g., an innovative use of online boosting).